# Are weak or negative clinical recommendations associated with higher geographical variation in utilisation than strong or positive recommendations? Cross-sectional study of 24 healthcare services

Agne Ulyte ,[1] Wenjia Wei,[1] Oliver Gruebner,[1,2] Caroline Bähler,[1,3] Beat Brüngger ,[1,3] Eva Blozik,[3,4] Viktor von Wyl,[1] M Schwenkglenks,[1] Holger Dressel[5]

► Prepublication history and supplemental material for this paper is available online. To view these files, please visit the journal online (http://dx.doi.org/10.1136/bmjopen-2020-044090).

For numbered affiliations see end of article.

**Correspondence to**
Dr Agne Ulyte;
agne.ulyte@uzh.ch

## ABSTRACT

**Objectives** When research evidence is lacking, patient and provider preferences, expected to vary geographically, might have a stronger role in clinical decisions. We investigated whether the strength or the direction of recommendation is associated with the degree of geographic variation in utilisation.

**Design** In this cross-sectional study, we selected 24 services following a comprehensive approach. The strength and direction of recommendations were assessed in duplicate. Multilevel models were used to adjust for demographic and clinical characteristics and estimate unwarranted variation.

**Setting** Observational study of claims to mandatory health insurance in Switzerland in 2014.

**Participants** Enrolees eligible for the 24 healthcare services.

**Primary outcome measures** The variances of regional random effects, also expressed as median odds ratios (MOR). Services grouped by strength and direction of recommendations were compared with Welch's t-test.

**Results** The sizes of the eligible populations ranged from 1992 to 409 960 patients. MOR ranged between 1.13 for aspirin in secondary prevention of myocardial infarction to 1.68 for minor surgical procedures performed in inpatient instead of outpatient settings. Services with weak recommendations had a negligibly higher variance and MOR (difference in means (95% CI) 0.03 (−0.06 to 0.11) and 0.05 (−0.11 to 0.21), respectively) compared with strong recommendations. Services with negative recommendations had a slightly higher variance and MOR (difference in means (95% CI) 0.07 (−0.03 to 0.18) and 0.14 (−0.06 to 0.34), respectively) compared with positive recommendations.

**Conclusions** In this exploratory study, the geographical variation in the utilisation of services associated with strong vs weak and negative vs positive recommendations was not substantially different, although the difference was somewhat larger for negative vs positive recommendations. The relationships between the strength

## Strengths and limitations of this study

► Although the strength and direction of recommendations is generally expected to influence the variation in clinical decisions, this is the first study to analyse this relationship quantitatively.
► The effect of the strength and direction of a recommendation on the geographical variation in healthcare utilisation was assessed within a comprehensive set of 24 healthcare services.
► Unwarranted variation of the services utilisation was extracted with a single standard approach.
► Indirect relationship and modifiers of the effect could not be studied.

or direction of recommendations and the variation may be indirect or modified by other characteristics of services. As initiatives discouraging low-value care are gaining attention worldwide, these findings may inform future research in this area.

## BACKGROUND

According to the evidence-based medicine (EBM) framework, clinical decisions should be guided by research evidence, clinical circumstances, and patient preferences and be integrated with clinical expertise.[1] If evidence is weak or lacking, patient preferences and clinical expertise have a particularly strong role in the decision.[2 3] In a clinical practice guideline, such a situation would be reflected by a weak recommendation.[2] As patient preferences tend to vary geographically,[4] and physician practice styles are also significantly influenced by the region of practice,[5 6] clinical decisions associated with less

conclusive research evidence or weak recommendations may have higher geographical variation.

Surprisingly, there is little direct evidence whether weak recommendations are, in fact, associated with higher variation. The few available studies focused on a single specialty and did not quantify the variation in a uniform way, complicating the comparison of results.[7][8] Therefore, despite many studies highlighting the substantial geographic variation in the utilisation of various healthcare services,[9–11] it is not clear, what role the components of the EBM framework play for this variation.

A second potential contributor to different services having different degrees of geographic variation is the direction of recommendation: positive (prescriptive) or negative (proscriptive), as in Choosing Wisely recommendations.[12] Negative recommendations usually concern long-used low-value practices and are based on the lack of supporting evidence or evidence of harms.[12–14] In contrast to positive recommendations, which often introduce new services or indications, negative recommendations usually challenge existing practices that are justified primarily by clinical expertise and judgement, expected to vary regionally.[4] Positive and negative recommendations have different perceived barriers to their implementation,[15] which could contribute to different variation patterns as well. However, no study has directly compared the geographic variation associated with positive and negative recommendations.

The primary aim of this study was to assess whether healthcare services with weak recommendations are associated with higher geographical variation in utilisation. In addition, a secondary aim was to test the association of geographic variation with the direction of the recommendation.

## METHODS
### Study hypotheses
Although this is primarily an explorative study, we formulated two specific hypotheses. The primary hypothesis of the study was that healthcare services with weaker evidence, as reflected in weak recommendations in clinical guidelines, would have higher geographical variation in utilisation than those with strong recommendations. The secondary hypothesis was that services with negative (proscriptive) recommendations would have higher geographic variation compared with those with positive (prescriptive) recommendations.

### Selection of studied healthcare services
This study was part of a project assessing the geographic variation of the utilisation of a set of healthcare services in Switzerland.[16] Studied healthcare services were translated from selected recommendation statements in clinical practice guidelines, following a systematic approach. We collected clinical practice guidelines of Swiss, European and applicable international medical societies, used in Switzerland and guiding the care for major non-communicable diseases

(as defined by the Swiss Federal Office of Public Health[17]). Recommendation statements from selected clinical practice guidelines were considered pragmatically by the authors according to their clinical relevance, the expected frequency of service use, and the size of the eligible population. Identified recommended or discouraged services were then screened for feasibility of measuring the utilisation in eligible populations with Swiss health insurance claims data, based on an approach described earlier.[18]

We aimed for the selected services to reflect both strong and weak, positive and negative recommendations, as well as different healthcare services types. We focused particularly on outpatient primary healthcare services, as they are relevant to the biggest part of the population. However, we also included some discouraged services outside primary healthcare to extend the spectrum of populations investigated.

The final selection comprised 24 services, including services for screening (N=4), diagnosis (N=6), primary prevention (N=1), treatment (N=4) and secondary prevention (N=9). Definitions of the selected services are provided in online supplemental file 1.

### Assessment of recommendations: strength and direction
Once the services were selected, their associated recommendations were formally assessed.

For each service, we selected the guideline in which the service was originally identified, and also looked up corresponding guidelines by the relevant European, American and international clinical societies. From this set of guidelines, we selected for assessment the one that was the most applicable to Switzerland in 2014 (see online supplemental file 2 for the prioritisation algorithm). We did not consider the guidelines of Swiss medical societies, as their quality of reporting is partially low,[19] and they tend to be consistent with European and international guidelines. If the service was initially selected based on a guideline published after 2014, and no applicable guideline could be identified for 2014, the recommendation was automatically considered weak.

Thus, a single recommendation statement was assessed for each service. The assessment was done in duplicate by two authors (AU and HD, both medical doctors). Discordant judgements were resolved with mutual agreement in a discussion. Each recommendation was classified as strong or weak (corresponding to Grading of Recommendations Assessment, Development and Evaluation (GRADE) definition[20]), and positive or negative. The algorithm and criteria for the classification are detailed in online supplemental file 2 for the strength, and in online supplemental file 3 for the direction of a recommendation. The list of guidelines containing the recommendation statements that were assessed is provided in online supplemental file 4.

### Swiss health insurance claims data
The utilisation of the selected healthcare services was evaluated using mandatory health insurance claims data from

the Helsana Group, covering approximately 1.2 million people (15% of the Swiss population). Helsana Group is one of several private companies providing mandatory health insurance in Switzerland. Eligible patient populations were identified from the patients enrolled with Helsana in 2014. Patients with incomplete address information, living in nursing homes and receiving reimbursement via lump-sums (masking some outpatient services), asylum seekers, those living outside Switzerland, and Helsana employees were excluded. The data provided by Helsana were anonymised.

## Models of geographic variation

The utilisation of each healthcare service was determined for each member of the eligible population (see online supplemental file 1 for definitions of the populations and services). For each service, the resulting binary outcome variable was modelled with a multilevel logistic regression technique, using 106 Swiss MobSpat regions ('mobilité spatiale'), as defined by the Swiss Federal Statistical Office,[21] as the higher level. MobSpat regions are constructed by combining several neighbouring municipalities based on geographic and population mobility criteria, and are often used as intermediate-size units of analysis for scientific and regional policy purposes. Each study participant's residence was assigned to the corresponding MobSpat region.

Fixed effects were estimated for the following explanatory variables: age, sex, number of comorbidities (0, 1, 2 and 3 or more), and clinical characteristics of relevance for specific indicators (see online supplemental file 1). These variables are often viewed as associated with warranted variation.[22] In contrast, we did not adjust for variables associated with unwarranted variation (eg, insurance characteristics or provider density). From each multilevel model, we extracted the variance of the regional random effects, reflecting the potentially unwarranted geographic variation. We also converted the variance to median odds ratios (MORs) for more convenient interpretation[23 24] and plotting. MOR is interpreted as the median odds of service utilisation by two individuals with identical characteristics in two randomly selected regions. As MOR is directly extrapolated from the variance, the ranking of these two parameters coincides.

## Statistical analysis of the hypotheses

Variances of the regional random effects of services utilisation from the models were used as data points in the final analysis of the hypotheses. Variances of services associated with weak and strong recommendations, as well as negative and positive recommendations were compared with Welch's unequal variances t-test. Mean differences and 95% CIs were presented. The same analysis was also performed for MOR, to improve interpretability of detected group differences.

Although the number of the services analysed was rather small (24), the distribution of the analysed variances was deemed sufficiently close to normal to warrant the use of parametric tests. To account for the small and unequal sample sizes, we used Welch's t-test, which is considered more robust in this setting.[25] CIs were not adjusted for multiple testing.

Statistical analyses were performed using R V.3.6.0[26] and MLwiN V.3.01[27] integrated in STATA V.14.2 using the runmlwin package.[28]

## Patient and public involvement

This study was performed as part of the National Research Programme 74 'Smarter Healthcare' of the Swiss National Science Foundations. Patients and public, including policy-makers and healthcare services providers, are involved in interpreting, disseminating and translating the overall results of studies conducted under this programme. Representatives of patients, healthcare providers, health insurers and healthcare policy-makers are members of the advisory board of the project. They provided feedback on the planned study design and its preliminary results. Individual patients were not directly involved in the planning and conducting of this study.

## RESULTS

Characteristics of the eligible populations and the geographic variation of the services are shown in table 1. Across the services, the sizes of the eligible populations ranged from 1 992 patients with a new disease-modifying antirheumatic drug prescription to 409 960 patients with recommended influenza vaccination. MOR, reflecting potentially unwarranted geographic variation in utilisation of the services, ranged from 1.13 (1.02–1.29) for aspirin in secondary prevention of myocardial infarction (MI) to 1.68 (1.53–1.87) for minor surgical procedures performed in inpatient instead of outpatient settings.

For three services, a major guideline relevant in 2014 in Switzerland could not be identified (long-term use of proton pump inhibitors, minor inpatient surgery procedures, elective Caesarean section). A total of eight services had weak, and six services had negative underlying recommendations. MOR was 1.29 for services with weak and 1.25 for services with strong recommendations; 1.26 for services with positive and 1.46 for services with negative recommendations (figure 1).

Based on Welch's t-test, the difference in mean variances (95% CI) of services with weak and strong recommendations was 0.03 (−0.06 to 0.11), and the difference in mean MOR was 0.05 (−0.11 to 0.21). The difference in mean variances (95% CI) of services with negative and positive recommendations was 0.07 (−0.03 to 0.18) and the difference in mean MOR was 0.14 (−0.06 to 0.34).

## DISCUSSION

We did not find a direct association between the strength of clinical recommendation and the geographical variation in the utilisation of 24 healthcare services. The geographical variation in the utilisation of services with underlying negative recommendations was slightly higher

**Table 1** Characteristics of the recommended or discouraged healthcare services studied

| Category | Healthcare service (abbreviated) | Eligible population | | | | Recommendation | | Random effects in multilevel model | |
|---|---|---|---|---|---|---|---|---|---|
| | | Utilisation in eligible population | Total N | Mean age (SD) | Female N (%) | Strength | Direction | Variance | Median OR (MOR) |
| Screening | Colon cancer screening | 5.9% | 276387 | 58.6 (5.8) | 142675 (51.6) | Strong | Positive | 0.04 (0.03–0.06) | 1.21 (1.17–1.26) |
| | Breast cancer screening | 20.9% | 178145 | 61.0 (7.2) | 178145 (100) | Weak | Positive | 0.22 (0.16–0.29) | 1.56 (1.47–1.67) |
| | Prostate cancer screening | 28.4% | 145874 | 59.1 (6.2) | 0 (0) | Weak | Positive | 0.07 (0.05–0.10) | 1.29 (1.25–1.35) |
| | Osteoporosis screening | 4.9% | 60812 | 72.6 (8.7) | 60812 (100) | Weak | Positive | 0.08 (0.04–0.13) | 1.31 (1.22–1.41) |
| Diagnosis | DM: HbA1c test | 69.6% | 49198 | 66.6 (13.0) | 22138 (45.0) | Strong | Positive | 0.17 (0.12–0.23) | 1.48 (1.40–1.58) |
| | DM: renal function test | 44.3% | 49198 | 66.6 (13.0) | 22138 (45.0) | Strong | Positive | 0.06 (0.04–0.09) | 1.27 (1.22–1.33) |
| | DM: LDL test | 44.3% | 33975 | 60.1 (11.2) | 13977 (41.2) | Strong | Positive | 0.13 (0.09–0.19) | 1.42 (1.34–1.51) |
| | DM: eye examination | 55.5% | 49198 | 66.6 (13.0) | 22138 (45.0) | Weak | Positive | 0.07 (0.05–0.10) | 1.29 (1.24–1.35) |
| | **TSH screening** | 76.1% | 169232 | 56.8 (18.5) | 111847 (66.1) | Strong | Negative | 0.18 (0.13–0.25) | 1.50 (1.42–1.61) |
| | **POCR** | 13.0% | 47215 | 60.3 (17.2) | 27086 (57.4) | Strong | Negative | 0.18 (0.13–0.26) | 1.50 (1.40–1.62) |
| Primary prevention | Influenza vaccination | 20.9% | 409960 | 64.1 (16.3) | 230202 (56.2) | Strong | Positive | 0.04 (0.03–0.05) | 1.20 (1.17–1.24) |
| Treatment | **Benzodiazepines** | 18.6% | 243951 | 75.0 (7.6) | 141986 (58.2) | Strong | Negative | 0.14 (0.10–0.18) | 1.42 (1.36–1.50) |
| | **Proton pump inhibitors** | 55.5% | 153523 | 55.7 (17.8) | 93543 (60.9) | Weak | Negative | 0.02 (0.02–0.03) | 1.16 (1.13–1.19) |
| | **Inpatient procedures** | 61.4% | 10656 | 50.5 (13.7) | 7719 (72.4) | Weak | Negative | 0.30 (0.20–0.43) | 1.68 (1.53–1.87) |
| | **Caesarean section** | 28.5% | 9449 | 31.9 (5.1) | 9449 (100) | Weak | Negative | 0.05 (0.02–0.09) | 1.24 (1.16–1.34) |
| Secondary prevention | AMI: aspirin | 47.0% | 2232 | 72.4 (13.7) | 801 (35.9) | Strong | Positive | 0.02 (0.00–0.07) | 1.13 (1.02–1.29) |
| | AMI: statin | 34.2% | 2232 | 72.4 (13.7) | 801 (35.9) | Strong | Positive | 0.14 (0.06–0.27) | 1.43 (1.25–1.63) |
| | AMI: beta-blocker | 42.1% | 2232 | 72.4 (13.7) | 801 (35.9) | Strong | Positive | 0.05 (0.00–0.13) | 1.25 (1.05–1.40) |
| | AMI: ACE/ARB | 43.8% | 2232 | 72.4 (13.7) | 801 (35.9) | Strong | Positive | 0.04 (0.00–0.12) | 1.21 (1.03–1.39) |
| | AMI: P2Y12 inhibitors | 46.8% | 2232 | 72.4 (13.7) | 801 (35.9) | Strong | Positive | 0.03 (0.00–0.10) | 1.18 (1.04–1.36) |
| | PPI with NSAID | 43.5% | 95072 | 61.0 (16.2) | 60804 (64.0) | Strong | Positive | 0.02 (0.01–0.03) | 1.15 (1.12–1.18) |
| | PAD: statin | 28.5% | 23868 | 63.6 (16.5) | 12113 (50.7) | Strong | Positive | 0.04 (0.03–0.07) | 1.22 (1.17–1.28) |
| | Afib: anticoagulation | 27.5% | 8291 | 80.8 (7.9) | 4037 (48.7) | Strong | Positive | 0.05 (0.02–0.09) | 1.24 (1.16–1.33) |
| | GCC with new DMARD | 58.7% | 1992 | 59.2 (15.3) | 1369 (68.7) | Weak | Positive | 0.06 (0.01–0.18) | 1.27 (1.07–1.49) |

Healthcare services, highlighted in bold, are associated with a negative recommendation. Utilisation was assessed within 1 year, 2014, including for services that are recommended less frequently (eg, colon cancer screening).

Afib, atrial fibrillation; AMI, acute myocardial infarction; ARB, angiotensin II receptor blockers; DM, diabetes mellitus; DMARD, disease-modifying antirheumatic drug; GCC, glucocorticosteroid drugs; HbA1c, glycated haemoglobin; LDL, low density lipid; NSAID, non-steroidal anti-inflammatory drugs; PAD, peripheral artery disease; POCR, preoperative chest radiography; PPI, proton pump inhibitors; TSH, thyroid-stimulating hormone.

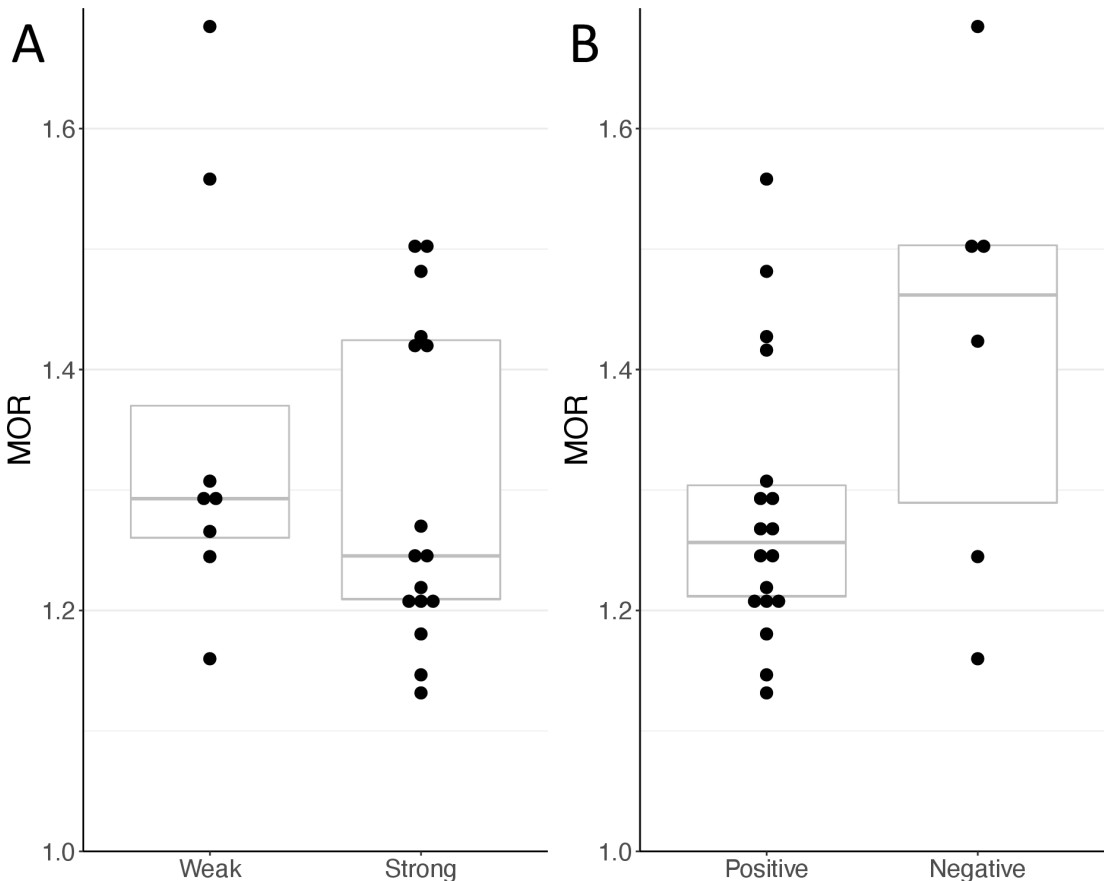

**Figure 1** Geographical variation of the healthcare services grouped by strength (A) and direction (B) of recommendations. (A) weak and strong recommendations; (B) positive and negative recommendations. Boxplots depict the interquartile range values (upper and lower hinges) and the median value. MOR, median OR.

than for those with positive recommendations, and for services with underlying weak recommendations than for those with strong recommendations. The difference was larger for negative versus positive recommendations; however, both differences were not statistically significant. In general, moderate potentially unwarranted geographical variation was observed, with MOR smaller than 1.50 for all but one service.

At least two other studies have to some extent examined the association between the strength of recommendations and the variation in adherence, each focusing on a single clinical specialty. In et al[7], examining a set of recommendations in oncology, found higher variation in the utilisation of services associated with a lower level of evidence. However, this study focused not on regional but on interinstitutional variation, comparing two groups of providers. In contrast to this study and in agreement with our results, Mayer et al[8] found that surgeon practices in knee and hip arthroplasty in Australia varied regardless of the strength of evidence available. Potentially, different clinical areas could be associated with different barriers to guideline implementation, modifying the relationship between recommendations and variation.

To better understand why a direct association of recommendation strength and variation in adherence was not observed, it may be useful to revisit the EBM framework.[1]

The EBM framework is normative and defines how clinical decisions *should* be made.[1 29] However, this may not always coincide with how decisions *are* made—a process analysed by descriptive theories.[29] In fact, the EBM model has been developed as conceptual rather than practical guidance of evidence implementation,[1] and has not yet generated a coherent theory of clinical decision making, and in particular, of how evidence is incorporated.[30] Thus, although a direct relationship between the strength of recommendation and the geographical variation of service utilisation would be encouraged by the normative EBM framework, it may not always be observed.

There are numerous reasons why even strong recommendations,[31 32] or conclusive research evidence more broadly,[33] may not directly translate into clinical practice. Research on knowledge translation has identified multiple barriers at different levels of the healthcare system, including structural, organisational, peer-group and professional factors[34] – many of which depend on the specific context where a service is provided, and thus may vary geographically. Knowledge transfer processes are highly non-linear and rely on triggering mechanisms.[35] Factors external to research evidence significantly affect translation—potentially creating large geographic heterogeneity even within services with strong recommendations. Finally, strong recommendations sometimes

describe care with varying patient preferences. For example, although colon cancer screening is strongly recommended, patient preferences for test attributes and modalities vary significantly.[36 37]

An influential framework, explaining different degrees of variation between healthcare services, has been proposed by Wennberg *et al*.[38] According to this framework, services are classified into effective, preference-sensitive, and supply-sensitive care. Effective care (services based on solid evidence, so that virtually all patients would choose them) largely corresponds to services with strong recommendations, as defined by GRADE and applied in this study.[2] Preference-sensitive care partly corresponds to services with weak recommendations, as they both imply trade-offs of risks and benefits of multiple options of care.[38] The utilisation of preference-sensitive surgical procedures usually has higher variation than of those associated with effective care.[39] In contrast, supply-sensitive care defines the frequency, setting and intensity of care provision rather than specific types of healthcare services. It is associated with high, supply-related variation, but is rarely discussed in guidelines,[39] and therefore, could not be included in our study. However, the service of minor surgeries performed as inpatient instead of outpatient procedures could be considered close to the supply-sensitive category. In fact, it had the highest MOR (1.68) in our study.

Regarding the secondary hypothesis, we found that services associated with negative recommendations had slightly higher geographic variation. We did not find other studies directly comparing the regional variation of services with the direction of recommendations. Few studies, focusing mostly on low-value care, have reported MOR as an expression of geographical variation, further limiting the comparison. For example, in a study by Badgery-Parker *et al*.[40] services discouraged by Choosing Wisely were shown to have regional MOR from 1.1 to 2.6—a range that includes all of our observed MORs.

Negative recommendations usually address a widespread service that lacks supporting evidence of benefit or the benefit is outweighed by harms.[2] In contrast to services with positive recommendations, which are introduced after supporting evidence is produced, services with negative recommendations typically become part of the clinical practice before evidence is sufficient to rule out their overall benefit. Therefore, their use could be related more to clinical expertise and practice, and could be expected to vary locally. Indeed, the barriers to implementing positive and negative recommendations seem to be different[15]—signalling that the pathways how they are interpreted and integrated into clinical decisions might also be different. As Choosing Wisely and similar initiatives are increasingly gaining attention,[41] our finding of higher geographic variation associated with negative recommendations may inform future research and implementation strategies.

This exploratory study has several limitations. First, although we aimed at a balanced selection of clinical fields and service types, the number (24) and range of studied services was limited by the data source, leading to somewhat unbalanced groups of strong and weak, positive and negative recommendations. Swiss claims data lack information on outpatient diagnoses, inpatient treatment details, and clinical information such as test results.[18] Lack of clinical information also meant that some populations were not as specific as defined by the recommendation. For example, beta-blockers and ACE inhibitors are recommended after an MI contingent on heart failure and left ventricular dysfunction. As such clinical details were unavailable, we had to rely on them being present in the majority of the hospitalised MI cases and distributed equally geographically. However, we believe that estimates of variation are accurate, as each of the 24 data points was generated by multilevel modelling of utilisation in populations of 85 000 patients on average, including all major explanatory variables such as age, sex and indicators of morbidity. Second, the services studied were unavoidably different by characteristics other than the strength or direction of recommendation, such as service type or clinical area, potentially resulting in confounding. Indeed, although distributed among all recommendation types, diagnostic services had somewhat higher regional variation in utilisation compared with treatment services (see online supplemental file 5). Although most of the selected services are delivered by primary care providers, their varied nature also means that the applied MobSpat regional units might not capture the regional variation equally well. Third, both the observed utilisation and its geographical variation depend on the definition of the service and population.[42] We aimed to measure the unwarranted variation in utilisation by using service-specific denominators (eligible populations) and adjusting for relevant clinical characteristics. How exactly unwarranted and warranted variation should be defined and measured, and what adjustments are necessary to differentiate them, is debated.[22 43] Fourth, the grouping of recommendations by strength and direction was partly subjective, although we tried to make it reproducible with a clear algorithm, implemented in duplicate. Unfortunately, many different systems for evaluating the strength of recommendations exist,[44] which cannot be easily reconciled, and the most prominent, GRADE approach, is not always explicitly used.

To explore the studied questions further, the sample of services could be expanded to inpatient and specialist care. Further, a meta-study of the numerous individual studies of geographic variation in healthcare services could be undertaken. However, this would currently be challenging, as studies choose different adjustment variables and specificity of studied populations, and report the variation in different quantitative forms (eg, MOR, systematic component of variation, range). Furthermore, there is a need for qualitative studies of the reasons for the variability of clinical decisions and how clinical expertise in these decisions interacts with evidence, clinical circumstances and patient preferences. Qualitative

evidence could help to generate more complex hypotheses for further quantitative studies, built on a finer understanding of all the factors influencing the variability of clinical decisions. Specialty-specific sets of services could also be further investigated.

## CONCLUSIONS

In this exploratory study of 24 healthcare services mostly in the outpatient primary care setting, we did not observe a significant difference in the degree of geographic variation in utilisation of services associated with strong versus weak recommendations. Services associated with negative recommendations had slightly, although also not statistically significantly, higher geographical variation. The relationship between the strength of recommendations and the variation may be indirect or modified by other characteristics of services, such as service type or clinical area. As initiatives discouraging low-value care are gaining attention worldwide, these findings may inform future research in this area.

**Author affiliations**
[1]Department of Epidemiology, Epidemiology, Biostatistics & Prevention Institute, University of Zurich, Zurich, Switzerland
[2]Department of Geography, University of Zurich, Zurich, Switzerland
[3]Department of Health Sciences, Helsana Group, Zurich, Switzerland
[4]Institute of Primary Care, University of Zurich and University Hospital Zurich, Zurich, Switzerland
[5]Division of Occupational and Environmental Medicine, Department of Epidemiology, Epidemiology, Biostatistics & Prevention Institute, University of Zurich and University Hospital Zurich, Zurich, Switzerland

**Contributors** MS, VvW and HD developed the underlying study program. AU and HD developed study design, with support from all authors. AU, WW, CB, EB, BB did data extraction, preparation and management. WW, MS and OG developed multilevel models applied in this study. AU and WW analysed the data. AU drafted the manuscript, with major inputs from HD and MS, and contributions from all authors. All authors read and approved the final manuscript.

**Funding** This work was supported by the Swiss National Science Foundation (SNSF) National Research Program 'Smarter Health Care' (NRP 74), as part of project number 26, grant number 407440_167349.

**Competing interests** MS declares a grant from Helsana Insurance Group, outside the submitted work. Helsana Group provided support in the form of salaries for authors BB, EB and CB, but did not have any additional role in the study design, data collection and analysis, decision to publish, or preparation of the manuscript. The other authors declare no competing interests.

**Patient consent for publication** Not required.

**Ethics approval** According to the national ethical and legal regulations, ethical approval was not needed for this analysis of anonymised data. This was confirmed by a waiver of the competent ethics committee (Kantonale Ethikkommission Zürich, dated January 11, 2017, BASEC-Nr. Req-2017–00011).

**Provenance and peer review** Not commissioned; externally peer reviewed.

**Data availability statement** Data may be obtained from a third party and are not publicly available. The data underlying this study cannot be shared publicly because they are the property of Helsana (https://www.helsana.ch/en/helsana-group), and have restricted public access on grounds of patient privacy. The data are managed by Helsana and subsets of the database are available for researchers after request and under specific conditions. Data are available from Helsana ( gesundheitskompetenz@helsana.ch) for researchers who meet the criteria for access to confidential data. Helsana will consider the possibilities of the research proposal and decide to grant access if the research questions can be answered with use of the Helsana data. When requests are granted, data are accessible only in a secure environment.

**ORCID iDs**
Agne Ulyte http://orcid.org/0000-0001-7419-9778
Beat Brüngger http://orcid.org/0000-0001-6173-5375

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
