## [Reviewer comments · BMJ Open]

ARTICLE DETAILS

TITLE (PROVISIONAL)	Are weak or negative clinical recommendations associated with higher geographic variation in utilization than strong or positive recommendations? Cross-sectional study of 24 health care services
AUTHORS	Ulyte, Agne; Wei, Wenjia; Gruebner, Oliver; Bähler, Caroline; Brünger, Beat; Blozik, Eva; von Wyl, Viktor; Schwenkglenks, M; Dressel, Holger

VERSION 1 – REVIEW

REVIEWER	Amber Goedken College of Pharmacy University of Iowa Iowa City, Iowa United States
REVIEW RETURNED	03-Nov-2020

GENERAL COMMENTS	In this study, the association between strength (and direction) of recommendation and geographic variation in utilization is assessed. I believe there is a lot of valuable information in this manuscript, and I appreciate the authors undertaking this study. Noted strengths are the research was conducted ethically, the research question is clearly stated, the authors took great care to distinguish services with strong and weak (and positive/negative) recommendations, and the manuscript is generally well-written. Regarding weaknesses, I am concerned the current methods hamper the ability to answer the research question well, but I believe the methods can be strengthened to a point where the question can be answered well. Major comments Methods: geographic areas • I am surprised by the small amount of text dedicated to specifying how the geographic regions were selected. I really feel readers should be provided with much more detail about the process for determining geographic area boundaries since this is so relevant to the research question. How was the decision made to use these 106 regions versus some other division of geographic areas? Are these regions small enough to capture differences in patient preferences or physician practice styles? Can the same boundaries be used for all these diverse services? For example, if cardiologists are mainly prescribing medications post-myocardial infarction but primary care providers are driving cancer screenings, can the geographic areas used to capture differences in primary care provider practice style regarding screening also capture differences in cardiologist practice style for prescribing medications?o If use of the 106 regions is justified and they continue to be used, please provide more detail so readers can understand how the area
---

came to be divided into 106 regions.

Methods: balance between groups/selection of services

- I would have more confidence in the comparison between the strong and weak groups (this comment is true for positive versus negative comparison as well, but I'll say strong and weak for simplicity) if there was greater balance between the two groups. For two services to be included in the study, they should be comparable except for one having strong and one having weak recommendation, which I admit will be quite challenging. Essentially, I am asking for a service with a strong recommendation to be matched with a service with a weak recommendation. At a minimum, the two services should be similar in terms of the direction of the recommendation and the type of service (e.g. prevention, treatment, inpatient, outpatient, etc). There should also be some comparability/criteria for the percentage of utilization in the eligible population for each service. For example, I'm seeing higher utilization in the eligible population for a weakly recommended service (DM: eye examination) than a strongly recommended service (DM: renal function test). Some factor(s) is/are driving this difference, so it seems this percentage should be taken into consideration when selecting matched services. To achieve what I am asking, some of the currently included services may need to be dropped and replaced with different services. This selection process for services should be described in detail.

- o I know what I am asking is difficult to achieve, but as the methods are now, I'm lacking confidence in the difference in mean variance between the groups.

Methods: eligible population

- I am struggling a bit to clearly understand who is included in the "eligible population" for services that are not received every year. For example, colonoscopy is listed in supplementary file 1 as being recommended every 10 years. The eligible population listed for this service is anyone 50-69 years old, but a chunk of these individuals had a colonoscopy less than 10 years prior to 2014 and wouldn't be expected to get a colonoscopy in 2014. Is this why the % utilization in the eligible population for colonoscopy is so low? This should be taken into consideration when selecting a comparable service with a weak recommendation, as detailed in my "balance between groups" comment above.

- It would be difficult to reproduce this study with the current level of detail provided about how the eligible populations were selected. For example, the eligible population for PAD: statin is described as "Adult patients undergoing diagnostic or treatment procedures for PAD" but the age cutoff for adult is not defined, and no procedure names or procedure codes are provided.

Minor comments

Abstract: p. 3, lines 46-49. Sentence about weak recommendation could be strengthened by noting higher variance and MOR compared to strong recommendation. Sentence about negative recommendation could be strengthened by noting higher variance and MOR compared to positive recommendation.

Discussion: p. 12, lines 218-219. It is not clear to me why slightly higher variation in utilization for negative versus positive is noted but not noted for weak versus strong when confidence intervals for both comparisons contain zero. If going to mention it, I would prefer

	slightly higher be noted for both comparisons. As written, I'm concerned reader given the impression there's more to conclude about negative/positive than weak/strong. I have this same concern with the conclusion section of the abstract (p. 3, lines 51-53).
--	---

REVIEWER	Erika Moen Dartmouth College, USA
REVIEW RETURNED	18-Dec-2020

GENERAL COMMENTS	The study examined whether regional variation in health care services was associated with recommendation strength and/or direction. The paper is well written, introduces the research question clearly, and appropriate references are cited to provide sufficient motivation for the study. The authors did not find strong evidence that the strength or direction of the recommendation was driving regional variation based on bivariate statistical tests. Is "region" defined using MobSpat utilization-based or based on other geospatially defined boundaries? Many readers may not be familiar with MobSpat (myself included), so a more detailed description of what these regions represent will be helpful, considering they are the unit of analysis. Any limitations to note by defining region in this way? Are there alternative definitions of region that may be relevant to consider? The authors state they do not adjust for variables associated with unwarranted variation (e.g., insurance characteristics or provider density). However, if they want to specifically measure the unwarranted variation that may be due to unmeasured factors such as provider preference, adjusting for some other measurable variables associated with unwarranted variation may be considered worthwhile as a sensitivity analysis. A discussion of any services that are known to have strong patient preferences would be of interest. Was there a significant difference in variance or MOR based on the category of service? Consider distinguishing category of service using color or shape in Fig. 1.
--

VERSION 1 – AUTHOR RESPONSE

Reviewer: 1

Dr. Amber Goedken, University of Iowa

Comments to the Author:

In this study, the association between strength (and direction) of recommendation and geographic variation in utilization is assessed. I believe there is a lot of valuable information in this manuscript, and I appreciate the authors undertaking this study. Noted strengths are the research was conducted ethically, the research question is clearly stated, the authors took great care to distinguish services with strong and weak (and positive/negative) recommendations, and the manuscript is generally well-written. Regarding weaknesses, I am concerned the current methods hamper the ability to answer the research question well, but I believe the methods can be strengthened to a point where the question can be answered well.

Response: Thank you for your comment and in general for all your helpful suggestions. We stressed in the conclusion that the study is exploratory and is meant to inform further research questions rather than give definitive answers (abstract conclusion on lines 52-4, discussion lines 223-6). Although it may not be feasible to perfect the methods of this particular study, also due to resource constraints, we hope that it will serve as a step towards further and more detailed answers explaining the role of clinical practice guidelines in the variation of health care utilization.

Major comments

Methods: geographic areas

- I am surprised by the small amount of text dedicated to specifying how the geographic regions were selected. I really feel readers should be provided with much more detail about the process for determining geographic area boundaries since this is so relevant to the research question. How was the decision made to use these 106 regions versus some other division of geographic areas? Are these regions small enough to capture differences in patient preferences or physician practice styles? Can the same boundaries be used for all these diverse services? For example, if cardiologists are mainly prescribing medications post-myocardial infarction but primary care providers are driving cancer screenings, can the geographic areas used to capture differences in primary care provider practice style regarding screening also capture differences in cardiologist practice style for prescribing medications? o If use of the 106 regions is justified and they continue to be used, please provide more detail so readers can understand how the area came to be divided into 106 regions.

Response: Thank you for the comment and suggestions. 106 Mobilité Spatiale (MobSpat) regions are defined by the Swiss Federal Statistical Office [1]. The regions were defined in 1982, and most recently updated in 2005. They are often used as intermediate-size units of analysis for scientific and regional policy purposes. The regions are constructed from several neighboring municipalities (the smallest regional administrative unit in Switzerland, 2202 in 2020), based on the geographic, structural and population mobility criteria. They are often used in health services research in Switzerland (including in our previous studies, e.g., [2]). We have revised the description of MobSpat regions in lines 155-7.

In other health service studies in Switzerland, often the 26 cantons or even the 3 major language regions are used to explore regional variation in Switzerland (e.g.,[3]). However, we have observed significant intra-cantonal variation in health care utilization in previous studies [4,5], and service-specific hospital catchment areas are complicated to construct for non-inpatient services due to lacking nationally collected data. MedStat regions, primarily designed to report inpatient service statistics (large enough so that individual patients cannot be identified), would be the most suitable alternative that could be considered. However, small population size for some of the reported services would preclude their consistent use. Therefore, we report only MobSpat regions, as they can be consistently analyzed for all services.

As the reviewer pointed out, various health care services are analyzed in this study. We chose to apply a single system of regional classification for all of them, to increase the comparability of the derived measures of variation. Most of the 24 services we analyzed are provided by general practitioners or other primary care providers. Such choice comes with the drawback, pointed by the review, that a single system of regional units might capture the variation not equally well for different services. We acknowledged this limitation in the revised manuscript (lines 295-7). Unfortunately, due to time and resource limitations, we were not able to develop individualized regional areas for each service – an approach which could be otherwise applied as sensitivity analysis and potentially considered in future studies.

Methods: balance between groups/selection of services

- I would have more confidence in the comparison between the strong and weak groups (this comment is true for positive versus negative comparison as well, but I'll say strong and weak for simplicity) if there was greater balance between the two groups. For two services to be included in the study, they should be comparable except for one having strong and one having weak recommendation, which I admit will be quite challenging. Essentially, I am asking for a service with a strong recommendation to be matched with a service with a weak recommendation. At a minimum, the two services should be similar in terms of the direction of the recommendation and the type of service (e.g. prevention, treatment, inpatient, outpatient, etc). There should also be some comparability/criteria for the percentage of utilization in the eligible population for each service. For example, I'm seeing higher utilization in the eligible population for a weakly recommended service (DM: eye examination) than a strongly recommended service (DM: renal function test). Some factor(s) is/are driving this difference, so it seems this percentage should be taken into consideration when selecting matched services. To achieve what I am asking, some of the currently included services may need to be dropped and replaced with different services. This selection process for services should be described in detail.

o I know what I am asking is difficult to achieve, but as the methods are now, I'm lacking confidence in the difference in mean variance between the groups.

Response: Thank you for your comment. Indeed, the current selection of a set of services, which needed to be identifiable with the available data source, was challenging. Previously, we have showed that only a limited part of all recommendations in clinical practice guidelines can be studied with health insurance claims data in Switzerland [6]. This restriction in the choice of recommendations also meant that we could not introduce further inclusion criteria, such as absolute level of utilization, of services. We have highlighted the limitations of the data source in the discussion already (lines 283-9), and added a further comment to acknowledge the disbalance of services in the different categories in the revised manuscript (lines 282-3).

The reviewer's suggested design of matching pairs of comparable services that would only be different in terms of their recommendation strength would be very valuable to explore in further studies. That might require using a different data source, and developing a different methodology of selecting and defining services and their target populations than the one developed for and adopted in this study. Therefore, simply removing some of the services would probably not lead to the matched-services design suggested by the reviewer. We would welcome (and indeed would be interested in pursuing) such approach in potential further studies. The selection, definition and operationalization of the services constituted the major part of the project that led to this study. Numerous clinical guidelines were screened, recommendations prioritized and then screened for identifiability with the available claims data source. Unfortunately, we would not have the resources to repeat the process within the scope of the current project.

Methods: eligible population

- I am struggling a bit to clearly understand who is included in the "eligible population" for services that are not received every year. For example, colonoscopy is listed in supplementary file 1 as being recommended every 10 years. The eligible population listed for this service is anyone 50-69 years old, but a chunk of these individuals had a colonoscopy less than 10 years prior to 2014 and wouldn't be expected to get a colonoscopy in 2014. Is this why the % utilization in the eligible population for colonoscopy is so low? This should be taken into consideration when selecting a comparable service with a weak recommendation, as detailed in my "balance between groups" comment above.
- It would be difficult to reproduce this study with the current level of detail provided about how the eligible populations were selected. For example, the eligible population for PAD: statin is described as "Adult patients undergoing diagnostic or treatment procedures for PAD" but the age cutoff for adult is not defined, and no procedure names or procedure codes are provided.

Response: Thank you for the comment. Indeed, the utilization of all services was assessed within a single year (2014). Therefore, for services that are recommended less frequently than annually, the absolute utilization appears artificially low. We have added a corresponding clarification in the footnote of the revised Table 1 (lines 198-9). We also revised the Additional file 1 to define the eligible populations more precisely, and added specific clinical codes used for identification of the health care service.

Minor comments

Abstract: p. 3, lines 46-49. Sentence about weak recommendation could be strengthened by noting higher variance and MOR compared to strong recommendation. Sentence about negative recommendation could be strengthened by noting higher variance and MOR compared to positive recommendation.

Response: Thank you for the suggested. We have revised by stating clearly the comparison (to strong and positive recommendations, respectively) (lines 47-9).

Discussion: p. 12, lines 218-219. It is not clear to me why slightly higher variation in utilization for negative versus positive is noted but not noted for weak versus strong when confidence intervals for both comparisons contain zero. If going to mention it, I would prefer slightly higher be noted for both comparisons. As written, I'm concerned reader given the impression there's more to conclude about negative/positive than weak/strong. I have this same concern with the conclusion section of the abstract (p. 3, lines 51-53).

Response: Thank you for the comment. Both services associated with strong vs weak and negative vs positive recommendations did not have significantly different variance or MOR. As we provide the confidence intervals for both differences in the abstract, we believe that a comment on statistical significance would not be of high priority in the abstract given the allowable word count. We revised the conclusion in the Discussion (lines 223-6, 220-1) to note that for both weak vs strong and negative vs positive recommendations a slight statistically non-significant difference was observed. We further revised the Discussion to stress that both differences were not statistically significant (i.e. 95% confidence interval for the difference overlaps 0) (lines 225-6, 321).

However, the size of the difference is considerably greater for negative vs positive as compared to weak vs strong recommendations. Therefore, we think that the conclusion is not entirely the same for these two pairs of comparisons. We have revised the conclusion of the abstract to state directly that the difference was not significant in both comparators, but somewhat larger for negative vs positive recommendations (lines 51-4).

Reviewer: 2

Dr. Erika Moen, Dartmouth College Geisel School of Medicine

Comments to the Author:

The study examined whether regional variation in health care services was associated with recommendation strength and/or direction. The paper is well written, introduces the research question clearly, and appropriate references are cited to provide sufficient motivation for the study. The authors did not find strong evidence that the strength or direction of the recommendation was driving regional variation based on bivariate statistical tests.

Is "region" defined using MobSpat utilization-based or based on other geospatially defined boundaries? Many readers may not be familiar with MobSpat (myself included), so a more detailed description of what these regions represent will be helpful, considering they are the unit of analysis.

Any limitations to note by defining region in this way? Are there alternative definitions of region that may be relevant to consider?

Response: Thank you for the comment and all your valuable input. We have added further clarification on how MobSpat regions are constructed in the Methods (lines 155-7). The regions are constructed based on geographic (e.g., physical barriers between the regions) and population mobility criteria, by combining several smallest administrative units of Switzerland.

As noted by another reviewer, we applied the same regions to all services. Although most of the analyzed services are provided as part of primary care, potentially, the limitation of such approach is that potentially regional variation is not captured equally well for all services. We added this limitation to the Discussion (lines 295-7).

Alternatively, cantons, smaller MedStat regions, and service-specific hospital catchment areas have been used in health services research in Switzerland. However, often there is considerable intra-cantonal variation in Switzerland [5], and service-specific hospital catchment areas are complicated to construct for non-inpatient services due to lacking nationally collected data. MedStat regions, primarily designed to report inpatient service statistics (large enough so that individual patients cannot be identified), would be the most suitable alternative that could be considered. However, small population size for some of the reported services would preclude their consistent use. Therefore, we report only MobSpat regions, as they can be consistently analyzed for all services.

The authors state they do not adjust for variables associated with unwarranted variation (e.g., insurance characteristics or provider density). However, if they want to specifically measure the unwarranted variation that may be due to unmeasured factors such as provider preference, adjusting for some other measurable variables associated with unwarranted variation may be considered worthwhile as a sensitivity analysis.

Response: Thank you for your suggestion. Indeed, choosing the variables to adjust for is a complex question in health services research. Therefore, we tried to limit the number of variables we are adjusting for to only those associated with *warranted* variation (as described in Methods, lines 159-62), while also defining the eligible populations for health services based on the clinical recommendations as specifically as possible (to ensure that almost all patients would (or would not be) indicated for the service).

We have explored how further variables influence the regional variation in the utilization of these services in Switzerland in other studies. For example, we identified insurance characteristics as strong drivers of utilization [2]. Quantifying the different contributors to the variation, e.g., patient and provider preference, is beyond the scope of the current study, and while interesting, would need to be studied in further projects.

A discussion of any services that are known to have strong patient preferences would be of interest.

Response: Thank you for the comment. Preference-sensitive services often correspond to services with weak recommendations, as mentioned in the Discussion (lines 258-9). Ideally, only effective care, with little expected variation in patient preferences, should be strongly recommended. Colon cancer screening is an exception – although it is strongly recommended (for persons of certain age), the preferences of eligible patients vary significantly. We revised the Discussion to comment on this (lines 250-3).

Was there a significant difference in variance or MOR based on the category of service? Consider distinguishing category of service using color or shape in Fig. 1.

Response: Thank you for the question. Comparing all 5 analyzed service types would be limited due to small number of services in each category. Therefore, we grouped them into diagnostic (screening, diagnosis) and therapeutic (primary prevention, treatment, secondary prevention) groups. Figure 1 below shows the distribution of diagnostic and therapeutic services.

Figure 1 Distribution of MOR of diagnostic (red) and treatment (blue) services among services with weak vs strong (left) and positive vs negative (right) recommendations

Diagnostic and treatment services are well distributed among the weak/strong, and positive/negative recommendations. Based on Welch's t-test, the difference in mean variances [95CI%] of diagnostic and treatment services was 0.04 [-0.01, 0.11], and the difference in mean MOR was 0.11 [-0.01, 0.23]. Thus, although not statistically significant, the difference is larger than that for weak vs strong or negative vs positive recommendations. We have added a comment to the Discussion, among the limitations as a potential confounder (lines 294-5) and a Supplementary file 5 with the figure.

VERSION 2 – REVIEW

REVIEWER	Amber Goedken College of Pharmacy University of Iowa Iowa City, Iowa United States
REVIEW RETURNED	22-Feb-2021
GENERAL COMMENTS	Thank you to the authors for their thorough and thoughtful responses to my comments. I now better recognize the scope of the project described in the manuscript, and I wish the authors well as they continue their work in this area.